# Decreased Levels of SARS-CoV-2 Fusion-Inhibitory Antibodies in the Serum of Aged COVID-19 Patients

**DOI:** 10.3390/diagnostics12081813

**Published:** 2022-07-28

**Authors:** Astrid Malézieux-Picard, Fabien Abdul, François R. Herrmann, Aurélie Caillon, Pascale Ribaux, Yves Cambet, Sabine Yerly, Stéphanie Baggio, Nathalie Vernaz, Dina Zekry, Karl-Heinz Krause, Olivier Preynat-Seauve, Virginie Prendki

**Affiliations:** 1Division of Internal Medicine for the Aged, Department of Rehabilitation and Geriatrics, Geneva University Hospitals, 3 Chemin du Pont-Bochet, 1226 Thônex, Switzerland; dina.zekry@hcuge.ch (D.Z.); virginie.prendki@hcuge.ch (V.P.); 2Department of Microbiology and Molecular Medicine, Faculty of Medicine, University of Geneva, Rue Michel-Servet 1, 1206 Geneva, Switzerland; fabien.abdul@unige.ch; 3Division of Geriatrics and Rehabilitation, Department of Rehabilitation and Geriatrics, Geneva University Hospitals, 3 Chemin du Pont-Bochet, 1226 Thônex, Switzerland; francois.herrmann@hcuge.ch; 4Department of Pathology and Immunology, Faculty of Medicine, University of Geneva, Rue Michel-Servet 1, 1206 Geneva, Switzerland; aurelie.caillon@unige.ch (A.C.); pascale.ribaux@unige.ch (P.R.); karl.h.krause@hcuge.ch (K.-H.K.); 5Division of Laboratory Medicine, Geneva University Hospitals, 1205 Geneva, Switzerland; yves.cambet@unige.ch (Y.C.); sabine.yerly@hcuge.ch (S.Y.); 6Laboratory of Virology, Geneva Center for Emerging Viral Diseases, Geneva University Hospitals, 1205 Geneva, Switzerland; 7Division of Prison Health, Geneva University Hospitals, University of Geneva, 1206 Geneva, Switzerland; stephanie.baggio@hcuge.ch; 8Institute of Primary Health Care (BIHAM), University of Bern, 3012 Bern, Switzerland; 9Medical Directorate, Finance Directorate, Geneva University Hospitals, 1205 Geneva, Switzerland; nathalie.vernaz@hcuge.ch; 10Division of Infectious Diseases, Geneva University Hospitals, 1205 Geneva, Switzerland; 11Department of Diagnostics, Geneva University Hospitals, 1205 Geneva, Switzerland; olivier.preynat-seauve@hcuge.ch; 12Department of Medicine, Medical School, University of Geneva, 1205 Geneva, Switzerland

**Keywords:** SARS-CoV-2 fusion inhibitory antibody, elderly, frailty

## Abstract

Background: The SARS-CoV-2 pandemic was particularly devastating for elderly people, and the underlying mechanisms of the disease are still poorly understood. In this study, we investigated fusion inhibitory antibodies (fiAbs) in elderly and younger COVID-19 patients and analyzed predictive factors for their occurrence. Methods: Data and samples were collected in two cohorts of hospitalized patients. A fusion assay of SARS-CoV-2 spike-expressing cells with ACE2-expressing cells was used to quantify fiAbs in the serum of patients. Results: A total of 108 patients (52 elderly (mean age 85 ± 7 years); 56 young (mean age 52 ± 10 years)) were studied. The concentrations of fiAbs were lower in geriatric patients, as evidenced at high serum dilutions (1/512). The association between fiAbs and anti-Spike Ig levels was weak (correlation coefficient < 0.3), but statistically significant. Variables associated with fusion were the delay between the onset of symptoms and testing (HR = −2.69; *p* < 0.001), clinical frailty scale (HR = 4.71; *p* = 0.035), and WHO severity score (HR = −6.01, *p* = 0.048). Conclusions: Elderly patients had lower fiAbs levels after COVID-19 infection. The decreased fiAbs levels were associated with frailty.

## 1. Introduction

Severe acute respiratory syndrome coronavirus-2 (SARS-CoV-2) was responsible for 5.5 million deaths worldwide in January 2022, particularly among vulnerable elderly people with associated co-morbidities [1,2,3,4]. Changes in the innate and adaptive immunity, namely immunosenescence, could explain why infectious diseases are more frequent in aging populations [5,6]. During viral infections such as influenza, humoral immunity is a major player in reducing the severity of the disease [7]. Understanding neutralizing responses is thus essential to determine the onset of humoral immunity and should be investigated in elderly patients with COVID-19. The detection of total immunoglobulin antibodies against Spike (S) protein (anti-S Ig) is the basis of most current serological tests but does not predict the functional activity of antibodies that specifically neutralize the Spike/ACE2 interaction and impair viral entry.

Only a fraction of antibodies functionally inhibit viral entry. Generally, these antibodies are referred to as neutralizing antibodies (nAbs). It has been shown that nAbs develop between the 2nd and 12th day after the onset of symptoms [8,9,10]. Some studies reported that not all patients develop nAbs [11,12]. Older age, male gender, and disease severity have been associated with higher levels of nAbs, but the roles of comorbidities and frailty have rarely been assessed [13,14,15]. The detection techniques for nAbs are complex and not standardized (use of bioinformatics tools, native virus, experimental pseudovirus expressing the Spike protein, or in many instances, looking only at protein–protein interaction).

The spike/ACE2 interaction not only mediates viral entry, but also fusion of SARS-CoV-2-infected cells with neighboring ACE2-expressing cells. This mechanism is probably of relevance for COVID-19 pathogenesis and for SARS-CoV-2 immune escape.

In this study, we used fusion-inhibitor antibodies (fiAbs), i.e., antibodies that inhibit the fusion of spike-expressing cells with ACE2-expressing cells [16]. We quantified fiAbs in geriatric patients hospitalized for COVID-19 and compared them with fiAbs found in a younger population (<65 years). Our secondary objectives were to: compare the fiAbs level with SARS-CoV-2 anti-S-Ig in both age groups, identify predictive factors for their development, and assess whether the fiAbs level is associated with intra-hospital mortality.

## 2. Materials and Methods

### 2.1. Setting and Participants

The group of patients aged >65 years were recruited among the GEROCOVID cohort, a prospective, observational, single-center cohort, and included patients >65 years old admitted to acute medical wards in internal medicine or geriatrics for SARS-CoV-2 infection between 14 April 2020 and 7 May 2021 at the Geneva University Hospitals (HUG), Geneva, Switzerland. SARS-CoV-2 infection was diagnosed with a positive reverse transcriptase–polymerase chain reaction (RT-PCR) against SARS-CoV-2 in nasopharyngeal swabs (NPS). Elderly patients who did not have the capacity to consent were included with the consent of a trusted person. Patients with psychiatric disorders or for whom consent could not be obtained were excluded. Data and serum samples were collected at admission and discharge from the acute-care setting. Patients who had no sample at discharge were excluded (including deceased patients).

The group of patients aged <65 years were recruited among the STRAT-CoV cohort [17,18]. This study included patients hospitalized for COVID-19 at the HUG between 26 February and 31 May 2020, with an age <65 years; those who did not explicitly mention that they did not want to participate were included. Patients for whom sera were performed between 10 days and 3 weeks after the onset of symptoms and available at the virology lab were retrospectively included in this study. No patient was vaccinated at that time. Ethics approval was granted by the Cantonal Ethics Research Committee of Geneva (GEROCOVID: no. 2020-01248, STRAT-CoV: no. 2020-01070).

### 2.2. Recorded Data

Demographic data, comorbidities, the number of treatments, clinical frailty scale (CFS), primary symptoms onset (PSO), clinical findings at admission and during hospitalization, pneumonia severity score (Pneumonia Severity Index), COVID-19 severity score (WHO score), place of care (intensive care unit (ICU), intermediate care unit or acute unit), length of stay, and C-reactive protein (CRP) at admission and during hospitalization were recorded [19,20]. The WHO severity score is defined as follows: mild disease—symptomatic patients without evidence of viral pneumonia or hypoxia; moderate disease—pneumonia clinical signs with SpO2 ≥ 90% on room air; severe disease—pneumonia plus one of the following signs: saturation <90% on room air; respiratory rate >30 breaths/min; signs of severe respiratory distress (accessory muscle use, inability to complete full sentences; critical disease)—acute respiratory distress syndrome (ARDS), sepsis, septic shock, or other conditions that would normally require the provision of life-sustaining therapies such as mechanical ventilation (invasive or non-invasive) or vasopressor therapy [21]. CFS is a numerical scale that goes from 1 (very fit) to 9 (terminally ill) used and validated to evaluate frailty in elderly patients [19]. We categorized them as fit (score 1–3), vulnerable (score 4–6) and frail (score 7–9).

### 2.3. Detection of Total Anti-S Antibodies

Electrochemiluminescence immunoassay for the detection of total SARS-CoV-2 Ig antibodies in blood samples was performed using the Roche (Roche Diagnostics International Ltd., Rotkreuz, Switzerland) anti-SARS-CoV-2 Ig quantitative ECLIA kit [22]. The results were automatically reported as the analyte concentration of each sample in U/mL, with <0.80 U/mL interpreted as negative and ≥0.80 U/mL as positive for SARS-CoV-2 anti-S Ig antibodies.

### 2.4. Detection of fiAbs

An in vitro cell fusion assay was established between Hela cells stably transduced with the full-length S protein of SARS-CoV-2 and Hela cells stably transduced with the ACE2 receptor, both under the control of a ubiquitous promoter. When Hela-Spike and Hela-ACE2 are co-cultured, cellular fusion occurs through the Spike/ACE2 interaction. To quantify fusion, a dual split reporter system was introduced in Hela-Spike and Hela-ACE2 [16]. This reporter system emits luminescence when fusion is occurring.

The more the luminescence was detected, the more fusion occurred and the less fiAbs were present. To prevent variability between operators, assays, and reagents, an internal control made of pooled sera from SARS-CoV-2-negative patients was systematically included. The neutralization was always normalized for each serum dilution with this internal control and calculated as the ratio between the luminescence from the patient’s serum and the luminescence from the control serum.

To compare the neutralization between aged and control populations, 4 dilutions of sera were tested: 1/8, 1/32, 1/128, and 1/512. Three repeated measurements per person and per dilution were carried out (technical triplicates). With 3 repeated measurements, for 108 patients, we should have obtained 1296 measurements. However, due to minor and isolated technical problems, 1288 measurements were studied, and for each patient, between 9 and 12 assays were obtained.

### 2.5. Statistical Analysis

Participant’s characteristics were compared between the two groups with Fisher’s exact tests, unpaired *t*-tests, and Mann–Whitney U tests, as appropriate. Mixed-effect linear regression models were used to identify predictors of the luminescence intensity (dependent variable) with the group and dilution as the independent variables, in order to analyze repeated measures (4 dilutions and 3 measures per dilution) with missing values. The same models were also run with the group*dilution interaction terms. The same type of model was repeated for each dilution with only the group as the independent variable. Similar simple and multiple mixed-effects linear regression models were also run with the independent variables listed in Table 2. Coefficients of determination (R^2^) were reported. The significance level was set at *p* < 0.05. The threshold of clinical significance is generally 80%. A univariate logistic regression model was used to test the association between fiAbs and death. All the statistics were performed with the Stata statistical software, release 17.0 (StataCorp, College Station, TX, USA, 2021).

## 3. Results

### 3.1. Participants

In total, 108 patients were included in the study: 52 (48.1%) in the elderly and 56 in the young group. A total of 16 (23%) patients from the GEROCOVID cohort were excluded: 12 patients did not have any antibody testing at discharge (premature discharge, end of life, technical problem), and 4 patients had a negative PCR on NPS and COVID-19 diagnosis was not retained upon review of the medical record.

The mean age in each group was 85.3 ± 6.9 years and 52.3 ± 9.9 years, respectively. Sixty-six patients (61.1%) were male; 38.5% in the elderly and 82.1% in the young group. In the young group, all patients lived at home, whereas 46 (88.7%) in the elderly group did. Elderly patients were hospitalized earlier than young ones after the onset of symptoms (3.7 days ± 6.3 vs. 9.2 days ± 4.0). The average time from PSO to antibody testing was 15 ± 5 days for elderly and 13 ± 4 days for young patients. Clinical characteristics of patients are shown in Table 1. The main critical symptoms were cough (72.4%), dyspnea (67.3%), asthenia (41.0%), and diarrhea (25.7%). Among these symptoms present at admission, younger patients had a statistically higher prevalence of cough, myalgia, headache, dyspnea, anosmia, and ageusia. Elderly patients more often had cognitive disorders (17.3% vs. 1.8%), chronic cardiac failure (21.6% vs. 1.8%), kidney disease (17.3% vs. 3.6%), and lower BMIs (21.1 vs. 29.4). In the elderly group, the mean number of medications was 7.3 ± 4.1 vs. 2.0 ± 4.0 in the younger group. In this cohort, only seven patients were immunosuppressed. Elderly patients had a higher CFS score than younger and more comorbidities (*p* < 0.001): only 4 (7.7%) were fit vs. 53 (94.6%) unfit, 39 (75%) were vulnerable vs. 2 (3.6%) not, and 9 (7.3%) were frail vs. 1 (1.8%) strong. Young patients had more severe symptoms on admission. In our cohort, 16 patients (14.8%) had a mild case of COVID-19 (5 young vs. 11 elderly), 26 (24.1%) a moderate case (10 young vs. 16 elderly), 38 (34.3%) a severe case (19 young vs. 19 elderly), and 26 (24.1%) a critical one (22 young vs. 4 elderly). None of the included elderly patients were admitted to the ICU. The average number of days of fever was higher in the younger group (7.6 ± 8.7 vs. 6.4 ± 9.9; *p* < 0.001), but elderly patients had a longer length of stay than younger patients (31.5 ± 24.7 vs. 18.2 ± 17.8; *p* < 0.001).

### 3.2. Detection of SARS-CoV-2 fiAbs

For fiAbs detection, the Spike/ACE2-dependent fusion was quantified by luminescence emission (see Section 2). Through their capacity to impair such an interaction, fiAbs present in a patient’s serum reduced the fusion and, as a result, the luminescent signal.

Figure 1 showed that, upon addition of the respective serum dilutions, the mean fusion value was systematically higher in the older group. The value increased with the dilution, meaning that fewer SARS-CoV-2 fiAbs were detected.

Table 2 shows a mixed-effects linear model taking into account the repeated measures and the group*dilution interaction; the average fusion values were not different between the two groups (*p* = 0.327). There was a significant dilution effect (*p* < 0.001). We identified the presence of a dilution interaction between the groups, meaning that the effect was not symmetric, and we observed a group effect only for the 1/512 dilution (*p* = 0.019).

### 3.3. Comparison between fiAbs and Total Anti-S Ig Antibody Level

In the young group, the higher the dilution, the better the correlation (Figure 2). Indeed, the association between the SARS-CoV-2 fiAbs amount and anti-S Ig antibody level was 9.4% (*p* = 0.022) for 1/8 dilution, 15.7% (*p* = 0.003) for 1/32 dilution, 25.4% (*p* < 0.001) for 1/128 dilution, and 30.0% for 1/512 dilution (*p* < 0.001). In the elderly group, the association was not improved by the dilution.

### 3.4. Predictive Factors for the Development of SARS-CoV-2 fiAbs

The dilution, delay between PSO and antibody testing, CFS, and WHO severity scores were statistically associated with fiAbs levels (Table 3). The longer the time between the PSO and the antibody testing, the lower the fusion and the higher the fiAbs level. The level of fiAbs increased with clinical severity and decreased with frailty.

### 3.5. Factors Associated with Intra-Hospital Mortality

We observed a mortality of 12.3% of the same magnitude for both groups (*p* = 1). Using univariate logistic regression, the fiAbs level (evaluated by the luminescence) was unable to predict death (OR = 1.00 (0.99–1.01), *p* = 0.657). In our study, the WHO severity score was associated with intra-hospital death (OR = 9.03 (2.56–31.92), *p* < 0.001).

## 4. Discussion

This study reports that SARS-CoV-2 fiAbs levels were lower in elderly patients compared to younger patients who were hospitalized for COVID-19 during the first waves of the pandemic, through a new and original method for the detection of functionally relevant antibodies. SARS-CoV-2 fiAbs were only weakly correlated with total anti-S Ig antibody results in both groups of patients. Interestingly, fiAbs were lower in frail patients but increased in patients with a high severity score. SARS-CoV-2 fiAbs were not associated with intra-hospital mortality.

Studies on SARS-CoV-2 nAbs differ in terms of the population studied (age, ambulatory or hospitalized), the time of sampling (from the onset of the infection), and the detection methods, explaining frequent contradictory results. Despite the fact that elderly patients are more vulnerable and would benefit from additional data on the humoral response to COVID-19, very few studies have been conducted on this population.

### 4.1. fiAbs Level in the Elderly

How do our results detecting fiAbs compare to results using more conventional neutralizing antibody tests? Xu et al. reported that age was positively associated with a peak in neutralizing activity in patients infected with SARS-CoV-2 with a mean age of 51 years, but nAbs were associated with the severity of the disease [9]. In a cohort study with two age groups (<60 years and >80 years), Müller et al. reported that nAbs were lower in the elderly group after SARS-CoV-2 vaccination [23]. Taken together and compared with our observations, it is suggested that a decrease in nAbs/fiAbs is found in the geriatric population (i.e., >80 years) and, hence, is a feature of advanced immunosenescence.

### 4.2. Correlation between Neutralizing/Fusion-Inhibitory Abs and Anti-S Ig Levels

Some studies have addressed the correlation between nAbs and conventional SARS-CoV-2 serology. Huynh et al. described that higher titers of anti-S antibodies moderately correlated with higher titers of nAbs in patients who recovered from COVID-19; however, the correlation was only 51% [24]. Rockstroh et al. found a higher correlation in samples performed 6–9 months after the infection than in those taken 2 months later [25]. Our study found only a weak correlation between SARS-CoV-2 fiAbs and anti-S Ig levels. Even the best correlation found in our study (younger group; 1/512 dilution) was only 30%, which is statistically significant, but clinically not relevant. Indeed, for this type of correlation, 80% is considered a clinically relevant cut-off. To better explain this concept: with a correlation of 80%, in 8 out of 10 patients, the Ig levels reflect the neutralizing/fusion-inhibitory activity of the antibodies; with a correlation of 30%, the Ig levels reflect the neutralizing/fusion-inhibitory activity of the antibodies in only 3 out of 10 patients. Thus, early during the course of SARS-CoV-2 infection, there is a very poor correlation between functionally active antibodies and conventional serology. This correlation probably improves at several months after infection, possibly due to B cell maturation. However, from a clinical point of view, a test that becomes meaningful only several months after the disease is pointless for the clinical management.

### 4.3. Frailty and Severity: Factors Associated with Development of fiAbs

Geriatric COVID-19 patients generally develop a more severe disease than younger patients. In our study, we observed a paradox: frailty was associated with a decreased fiAbS response, while disease severity was associated with an increased fiAbs response. Finally, there was no association between the fiAbs response and mortality. Several hypotheses can explain this apparent paradox.

First, geriatric patients tend to have a more severe case of the disease with increased and prolonged exposure to the virus; one would expect an increased antibody response [26]. Thus, the observed antibody responses are determined by at least two divergent elements: increased antibody production because of more severe and prolonged disease and decreased antibody production because of immunosenescence. For example, Chen et al. showed that, among 49 patients (median age 37 years; IQR (30.0–54.5)) who recently recovered from COVID-19, those with higher titers of nAbs were older and had more extensive lung abnormalities and higher CRP [27]. After adjustment, only the severity of the disease and comorbidities correlated with nAbs titers at distance from the infection (43 days; IQR (36–50)).

Second, fiABs were measured about two weeks after the beginning of the onset of the primary symptoms, i.e., when the transition from limited upper respiratory infection to severe disease had in general already occurred. Thus, it would be interesting to investigate fusion inhibitory antibodies at an earlier stage, when fiAbs reflect only the immune response, but not yet the severity of the disease. Indeed, the delay of the humoral response may play a key role in the course of the disease. Kawasuji et al. showed that neutralization activity on admission was inversely correlated with disease severity, making the hypothesis that a rapid nAbs response may play an important role in preventing from severe disease [8]. In the same line, Lucas et al. demonstrated that the correlation of anti-S response with COVID-19 severity was time dependent [28]. Thus, there is most likely a critical time window in which nAbs should develop to improve the viral control and outcome.

Despite their vulnerability to COVID-19, elderly patients have been underrepresented in previous studies. In our study, frailty was systematically assessed, using the Rockwood frailty scale. We found that not age by itself, but rather frailty was associated with the level of fiAbs. Indeed, fiAbs levels were decreased in frailer patients. Such a hypothesis has been proposed by Vetrano et al. [29], but limited experimental support for this hypothesis exists. Thus, our study is among the first experimental evidence demonstrating that decreased generation of functional antibodies is closely linked to frailty.

Our study is important because it evaluates, for the first time, functionally active SARS-CoV-2 antibodies in the elderly, including co-morbidities and frailty. However, it has several limitations. First, there was a selection bias, as we could not assess all the patients of both cohorts. In our study population, patients with a very severe case of COVID-19 that died rapidly from the disease often could not be included in the study. This limits the generalizability of our results concerning the relationship between fiAbs levels and mortality. Second, it is a single-center study comparing two different cohorts and with a small sample size. Third, our samples were taken 10–15 days after the onset of symptoms; thus, they neither reflect the very early nor the long-term antibody response. Fourth, we did not assess SARS-CoV-2 variants. Fifth, we assessed neither inflammatory mediators nor T cell responses.

## 5. Conclusions

Post-COVID-19 serum of elderly patients had a significantly decreased capacity to block the fusion of SARS-CoV-2 spike with ACE-expressing cells, as revealed by studying high dilutions of the patient serum. The correlation between the commonly measured total anti-S Ig antibodies and the fusion-neutralizing Abs was statistically significant but weak. In other words, the level of total anti-Spike Ig antibodies does not allow us to predict the levels of fusion-neutralizing Abs in an individual patient. The levels of fiAbs were lower in frail patients, yet they were not associated with the risk of dying. Perhaps the number of patients in our study was too small to see an impact on mortality. However, an alternative explanation should be considered: higher fiAbs levels might not only be prospective indicators of protective immunity, but also indicators of the severity of the past disease. Thus, future studies on COVID-19 in elderly patients should prospectively investigate the protection provided by fiAbs (after vaccination or disease) and should also include an assessment of frailty. Such an approach should allow a better understanding of the immunological and clinical specificities of COVID-19 in the geriatric setting.

## Figures and Tables

**Figure 1 diagnostics-12-01813-f001:**
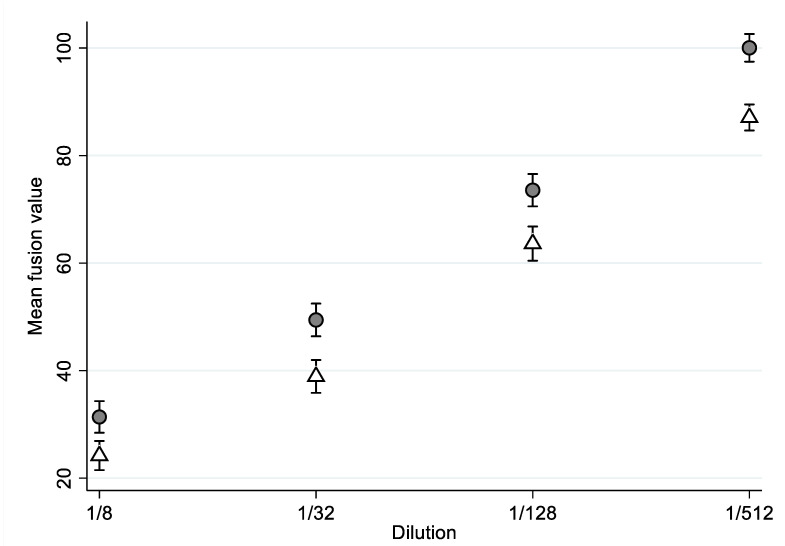
Mean fusion value by groups and by dilution. Young people are represented in white and elderly people in grey.

**Figure 2 diagnostics-12-01813-f002:**
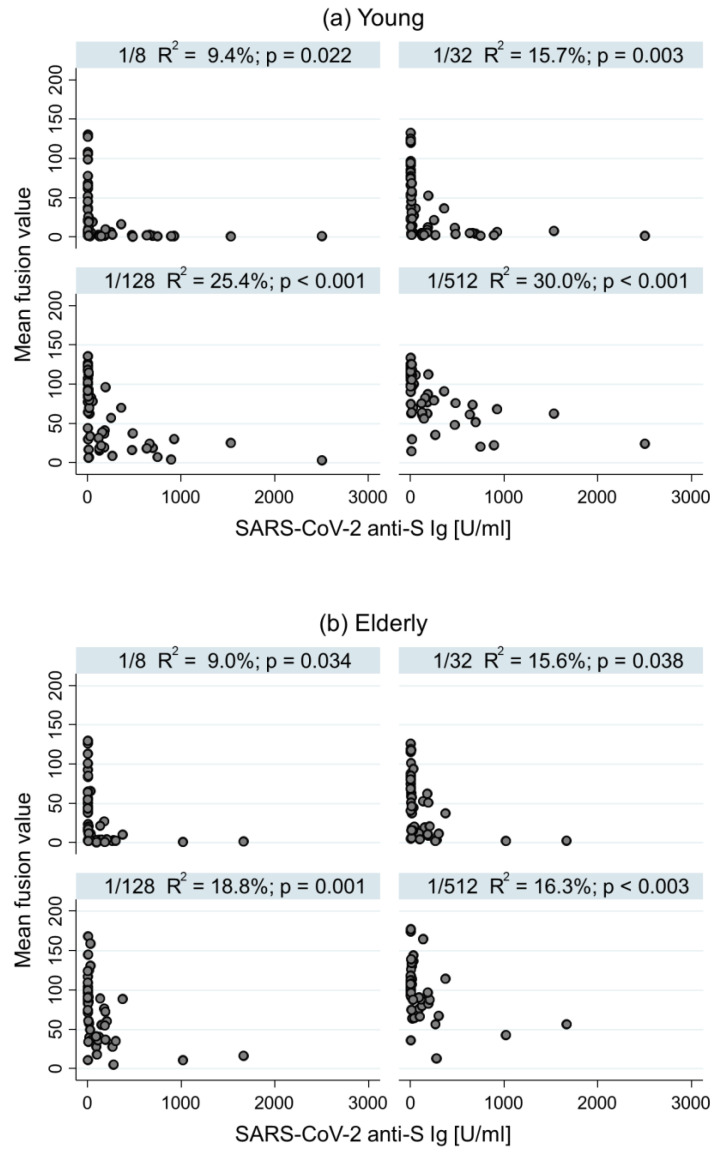
Association between fusion and anti-S Ig antibody level by groups and by dilution.

**Table 1 diagnostics-12-01813-t001:** Clinical and biological characteristics of study participants.

	Young	Elderly	Total	*p*-Value
	N; Mean ± SD	N; Mean ± SD	N; Mean ± SD	
Number	56	52	108	
Age (years)	56; 52.3 ± 9.9	52; 85.3 ± 6.9	108; 68.4 ± 18.6	<0.001
Gender (male)	46 (82.1%)	20 (38.5%)	66 (61.1%)	<0.001
Place of living (home)	56 (100.0%)	46 (88.7%)	102 (94.4%)	0.021
Number of comorbidities				<0.001
0	27 (48.2%)	0 (0.0%)	27 (25.0%)	
1	19 (33.9%)	46 (88.5%)	65 (60.2%)	
2	7 (12.5%)	2 (3.8%)	9 (8.3%)	
3	3 (5.4%)	2 (3.8%)	5 (4.6%)	
4	0 (0.0%)	2 (3.8%)	2 (1.9%)	
Clinical frailty scale				<0.001
1	36 (64.3%)	0 (0.0%)	36 (33.3%)	
2	11 (19.6%)	0 (0.0%)	11 (10.2%)	
3	6 (10.7%)	4 (7.7%)	10 (9.3%)	
4	1 (1.8%)	14 (26.9%)	15 (13.9%)	
5	1 (1.8%)	8 (15.4%)	9 (8.3%)	
6	0 (0.0%)	17 (32.7%)	17 (15.7%)	
7	1 (1.8%)	8 (15.4%)	9 (8.3%)	
8	0 (0.0%)	1 (1.9%)	1 (0.9%)	
9	0 (0.0%)	0 (0.0%)	0 (0.0%)	
COPD	0 (0.0%)	8 (13.5%)	8 (7.4%)	0.010
BMI	39; 29.4 ± 6.8	50; 24.1 ± 5.5	89; 26.4 ± 6.6	<0.001
Immunosuppression	4 (7.1%)	3 (5.8%)	7 (6.5%)	1.000
Diabetes	9 (16.1%)	8 (15.4%)	17 (15.7%)	1.000
Chronic cardiac failure	1 (1.8%)	11 (21.6%)	12 (11.2%)	0.001
Hypertension	12 (21.4%)	4 (7.7%)	16 (14.8%)	0.059
Kidney disease	2 (3.6%)	9 (17.3%)	11 (10.2%)	0.025
Cognitive disorders	1 (1.8%)	9 (17.3%)	10 (9.2%)	0.007
Active neoplasia-solid cancer or lymphoma	2 (3.6%)	3 (5.8%)	5 (4.6%)	0.670
Symptoms at admission (new or increasing)or biological markers				
Delay PSO—admission (days)	56; 9.2 ± 4.0	52; 3.7 ± 6.3	108; 6.5 ± 5.9	<0.001
Delay PSO—antibody testing (days)	56; 13.2 ± 3.9	52; 15.2 ± 5.5	108; 14.2 ± 4.8	0.030
Cough	46 (82.1%)	30 (61.2%)	76 (72.4%)	0.014
Sputum production	14 (25.0%)	7 (14.3%)	21 (20.0%)	0.155
Myalgia	23 (41.1%)	4 (8.2%)	27 (25.7%)	<0.001
Tiredness	20 (35.7%)	23 (46.9%)	43 (41.0%)	0.264
Delirium	4 (7.1%)	7 (14.3%)	11 (10.5%)	0.224
Headache	19 (33.9%)	4 (8.2%)	23 (21.9%)	0.001
Ageusia	8 (14.3%)	1 (2.0%)	9 (8.6%)	0.046
Anosmia	8 (14.3%)	0 (0.0%)	8 (7.6%)	0.006
Rhinorrhea	10 (17.9%)	4 (8.2%)	14 (13.3%)	0.139
Diarrhea	15 (26.8%)	12 (24.5%)	27 (25.7%)	0.583
Dyspnea	43 (76.8%)	27 (56.3%)	70 (67.3%)	0.036
Fall	1 (1.8%)	11 (22.4%)	12 (11.5%)	0.001
PSI	56; 75.7 ± 26.9	52; 11.4 ± 32.1	108; 44.8 ± 43.7	<0.001
C-reactive protein at admission (mg/L)	55; 101.9 ± 72.2	52; 50.2 ± 51.0	107; 76.7 ± 67.7	<0.001
Creatinine (µmol/L)	55; 105.1 ± 95.5	52; 87.8 ± 38.4	107; 96.5 ± 73.3	<0.001
During the hospitalization				
Respiratory distress syndrome				<0.001
No	15 (26.8%)	36 (69.2%)	51 (47.2%)	
At admission	31 (55.4%)	11 (21.2%)	42 (38.9%)	
During stay	10 (17.9%)	5 (9.6%)	15 (13.9%)	
Transfer to intermediate care unit	18 (34.6%)	7 (13.5%)	25 (24.0%)	0.021
Transfer to intensive care unit	22 (39.3%)	0 (0.0%)	22 (20.4%)	<0.001
Intra-hospital death	7 (12.7%)	6 (11.8%)	13 (12.3%)	1.000
Number of days with fever	54; 7.6 ± 8.7	51; 6.4 ± 9.9	105; 7.0 ± 9.3	0.029
Number of days with O_2_	56; 11.4 ± 10.9	50; 8.9 ± 11.1	106; 10.2 ± 11.0	<0.001
C-reactive protein higher value (mg/L)	55; 178.3 ± 113.9	43; 88.4 ± 63.9	98; 138.8 ± 104.9	<0.001
Length of hospital stay	56; 18.2 ± 17.8	51; 31.5 ± 24.7	107; 24.5 ± 22.2	<0.001
WHO severity score				0.001
Asymptomatic	0 (0.0%)	2 (3.8%)	(1.9%)	
Mild disease	5 (8.9%)	11 (21.2%)	16 (14.8%)	
Moderate disease	10 (17.9%)	16 (30.8%)	26 (24.1%)	
Severe disease	19 (33.9%)	19 (36.5%)	38 (35.2%)	
Critical disease	22 (39.3%)	4 (7.7%)	26 (24.1%)	

Abbreviations: BMI: body mass index, COPD: chronic obstructive pulmonary disease, PSI: Pneumonia Severity Index, PSO: primary symptoms onset.

**Table 2 diagnostics-12-01813-t002:** Interaction between groups and serum dilution. The interaction term is Group*dilution.

	Coefficient (%) (IC 95%)	*p*-Value
Elderly group	6.49 (−6.49–19.48)	0.327
Dilution		
1/32	14.71 (10.99–18.44)	<0.001
1/128	39.42 (35.70–43.15)	<0.001
1/512	62.88 (59.16–66.61)	<0.001
Group*dilution		
Elderly-1/32	4.02 (−1.39–9.42)	0.145
Elderly-1/128	3.78 (−1.63–9.20)	0.171
Elderly-1/512	6.47 (1.06–11.87)	0.019

**Table 3 diagnostics-12-01813-t003:** Factors associated with the development of SARS-CoV-2 fiAbs indirectly measured with fusion assessed with bivariate and multivariable multiple-mixed linear regressions.

	Bivariate Analysis		Multivariable Analysis	
	Coefficient (IC95%)	*p*-Value	Coefficient (IC95%)	*p*-Value
Dilution				
1/32	16.40 (13.68–19.12)	<0.001	16.40 (13.68–19.12)	<0.001
1/128	41.20 (38.49–43.91)	<0.001	40.98 (38.25–43.70)	<0.001
1/512	65.95 (63.25–68.66)	<0.001	66.02 (63.30–68.74)	<0.001
Age (years)	0.08 (−0.26–0.42)	0.470	−0.40 (−0.87–0.07)	0.092
Gender (male)	8.25 (−4.66–21.16)	0.210	2.20 (−10.76–15.16)	0.740
LOS	0.11 (−0.17–0.40)	0.436	0.06 (−0.26–0.38)	0.711
Delay PSO- antibody testing (d)	−2.41 (−3.64–−1.16)	<0.001	−2.69 (−3.85–−1.53)	<0.001
Arterial hypertension	−9.12 (−26.87–8.64)	0.313	−8.98 (−24.99–7.03)	0.272
Immunosuppression	9.14 (−16.54–34.82)	0.485	3.04 (−20.34–26.42)	0.799
Clinical Frailty Scale	2.70 (−0.16–5.57)	0.064	4.71 (0.34–9.10)	0.035
WHO severity score	−6.07 (−11.96–−0.18)	0.043	−6.01 (−11.97–−0.06)	0.048

LOS = length of stay, PSO = primary symptoms onset.

## Data Availability

Data, the statistical code, and technical processes are available from the corresponding author at astrid-marie.malezieux@hcuge.ch.

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
