# Peer review of "Decreased Levels of SARS-CoV-2 Fusion-Inhibitory Antibodies in the Serum of Aged COVID-19 Patients"

_diagnostics, 2022, doi:10.3390/diagnostics12081813_

Round 1
Reviewer 1 Report
I am honored to review the manuscript titled “the decreased levels of SARS-CoV-2 fusion-inhibitory antibodies in the serum of aged COVID-19 patients” for your esteemed journal MDPI. I regret to say that this study’s participants are very less.
The authors investigated the decreased levels of SARS-CoV-2 fusion-inhibitory antibodies in the serum of aged COVID-19 patients. The author’s study was designed on >65 years and <65 years old, patients were categorized into GEROCOVID cohort ( N=52) and STRAT-CoV cohort study (N=56) with a very small sample sizeand estimated the levels of fiAbs and SARS -CoV-2 anti -S Ig antibodies. The study is interesting, however, a few grammatical errors in some sentences which need to be corrected. Authors are requested to go through the manuscript and correct it grammatically.
Other comments
The study participants number (N) is 108 divided into elderly patients and younger patients (N=56)is too small for concluded your objectives. Especially, the discussion section is excessively irrelevant and did not properly discuss and support the findings (results) of the manuscripts. The results section was not clearly mentioned, readers were confused in its current form, and may be resubmitted after modifications.
I amsorry to informyou that your manuscript is not acceptable in its current form and may be resubmitted after modifications.
Author Response
We thank you for your comments on our article entitled “Decreased levels of SARS-CoV-2 fusion-inhibitory antibodies in the serum of aged COVID-19 patients.”
Here are our responses to each comment:
Point 1: I regret to say that this study’s participants are very less.
The authors investigated the decreased levels of SARS-CoV-2 fusion-inhibitory antibodies in the serum of aged COVID-19 patients. The author’s study was designed on >65 years and <65 years old, patients were categorized into GEROCOVID cohort ( N=52) and STRAT-CoV cohort study (N=56) with a very small sample sizeand estimated the levels of fiAbs and SARS -CoV-2 anti -S Ig antibodies. The study is interesting, however, a few grammatical errors in some sentences which need to be corrected. Authors are requested to go through the manuscript and correct it grammatically.
Response 1:
- You are absolutely right, the number of patients included in the study is relatively small. We have further detailed the limitations of the study (line 329-337).
- Following your comment, we used one of the editing services to check the grammar, spelling, ponctuation and phrasing of the manuscript.
Point 2: The study participants number (N) is 108 divided into elderly patients and younger patients (N=56)is too small for concluded your objectives. Especially, the discussion section is excessively irrelevant and did not properly discuss and support the findings (results) of the manuscripts. The results section was not clearly mentioned, readers were confused in its current form, and may be resubmitted after modifications.
Response 2:
- Your remark is pertinent, indeed the number of study participants is relatively small. We can, however, meet our main objective : quantify fiAbs in geriatric patients hospitalized for COVID-19 and compare with fiAbs found in a younger population (<65 years). We can also compare fiAbs level with SARS-CoV-2 anti-S-Ig in both age groups and identify predictive factors for their development. Because of the small number of events (deaths), there is a risk of overfitting when looking for predictors of in-hospital mortality with multivariable logistic regression. Thus, we removed Table 6.
- We tried to clarify the presentation of the results. In particular, we transformed Table 2 in Figure 1 (line 206) and Table 4 in Figure 2 (line 227).
- Thanks to your comments, we have made big changes. We have improved the discussion to make our message more understandable. We have redistributed the 3rd paragraph. We structured the discussion around the major points (lines 262, 273, 291). We have deleted the paragraph preceding the limitations because it discussed the relationship between frailty and mortality. We have also reformulated the limitations of the study.
Reviewer 2 Report
The paper presented for review “Decreased levels of SARS-CoV-2 fusion-inhibitory antibodies in the serum of aged COVID-19 patients” presents the levels of fiAbs in two research groups represented by two age categories. The methodology of the work and the results are very simple, although the authors presented and discussed them in a very ingenious way. Thanks to this, the work is very informative and definitely broadens our knowledge on such an important topic for us. The introduction is sufficient. Materials and methods are well described. Results presented correctly. Very good discussion with the use of appropriate literature. I believe the work should be published in Diagnostics.
Minor revisions:
1) I believe that some of the results from the tables could be presented in the form of graphs. This would increase the readability of the work and increase its aesthetic value.
2) Line 73 - "that" repeated
Author Response
We thank you for your comments on our article entitled “Decreased levels of SARS-CoV-2 fusion-inhibitory antibodies in the serum of aged COVID-19 patients.”
Here are our responses to each comment:
Point 1: I believe that some of the results from the tables could be presented in the form of graphs. This would increase the readability of the work and increase its aesthetic value.
Response 1: We completely agree with you. We tried to clarify the presentation of the results. In particular, we transformed Table 2 and Table 4 in Figures 1 (line 206) and 2 (line 227).
Point 2: Line 73 - "that" repeated
Response 2: Thank you for your comment. We removed the repetition. “This mechanism is probably of relevance for COVID-19 pathogenesis and for SARS-CoV-2 immune escape.”
Round 2
Reviewer 1 Report
accepted